

# Catalyzing rapid discovery of gold-precipitating bacterial lineages with university students

Noah G. Riley[1], Carlos C. Goller[1,2], Zakiya H. Leggett[3],
Danica M. Lewis[4], Karen Ciccone[4] and Robert R. Dunn[5,6,7]

[1] Department of Biological Sciences, North Carolina State University, Raleigh, NC, USA
[2] Biotechnology Program (BIT), North Carolina State University, Raleigh, NC, USA
[3] Department of Forestry and Environmental Resources (FER), North Carolina State University, Raleigh, NC, USA
[4] North Carolina State University Libraries, North Carolina State University, Raleigh, NC, USA
[5] Department of Applied Ecology, North Carolina State University, Raleigh, NC, USA
[6] University of Copenhagen, Natural History Museum of Denmark, Copenhagen, Denmark
[7] German Centre for Integrative Biodiversity Research (iDiv), Halle-Jena-Leipzig, Germany

Corresponding author
Carlos C. Goller, ccgoller@ncsu.edu

## ABSTRACT

Intriguing and potentially commercially useful microorganisms are found in our surroundings and new tools allow us to learn about their genetic potential and evolutionary history. Engaging students from different disciplines and courses in the search for microbes requires an exciting project with innovative but straightforward procedures and goals. Here we describe an interdisciplinary program to engage students from different courses in the sampling, identification and analysis of the DNA sequences of a unique yet common microbe, *Delftia* spp. A campus-wide challenge was created to identify the prevalence of this genus, able to precipitate gold, involving introductory level environmental and life science courses, upper-level advanced laboratory modules taken by undergraduate students (juniors and seniors), graduate students and staff from the campus. The number of participants involved allowed for extensive sampling while undergraduate researchers and students in lab-based courses participated in the sample processing and analyses, helping contextualize and solidify their learning of the molecular biology techniques. The results were shared at each step through publicly accessible websites and workshops. This model allows for the rapid discovery of *Delftia* presence and prevalence and is adaptable to different campuses and experimental questions.

## INTRODUCTION

The potential benefits from the study of the unique abilities of bacteria to everyday human life is ever more obvious. Bacteria are used industrially in food preparation, drug production, waste treatment and many other roles. Advances in biotechnology techniques have facilitated the use of known bacterial species and their enzymes, proteins and pathways (*Berini et al., 2017*). For example, it is now possible, and indeed not very difficult,

to identify genes of interest in a bacterial species, clip those genes out of that species and insert them into another work horse species of bacteria to allow the products of those genes to be produced industrially. Ironically, as our ability to harness the power of bacteria becomes ever more sophisticated, one of the key challenges is still finding the useful bacteria in the first place. In a world with as many as a trillion bacterial species (*Locey & Lennon, 2016*; *Pike, Viciani & Kumar, 2018*), how does one speed the discovery of bacterial species with a particular use or even simply strains of a particular bacterial taxon with sequences of interest?

One approach is to engage citizen scientists. In as much as the first step in the discovery of novel, useful microbes is often collection from nature, collections made by the public have the potential to speed up this key and often rate-limiting first step. What is more, in a rapidly interconnected digital era, the potential for truly global projects that rely on hundreds, thousands, or even hundreds of thousands of individuals is ever greater (*Cooper, 2016*). Citizen scientists contribute data to many publicly-accessible projects, from birdwatchers helping conservation efforts with the e-Bird project (https://ebird.org/home; *Sullivan et al., 2014*), game enthusiasts folding proteins for the FoldIt project (https://fold.it/portal/; *Cooper et al., 2010*), to homeowners exploring the microbial diversity in their houses (http://robdunnlab.com/projects/wild-life-of-our-homes/; *Dunn et al., 2013*). Additionally, projects like the Science Education Alliance—Phage Hunters Advancing Genomics and Evolutionary Science (SEA—PHAGES) and Tiny Earth engage students in large research projects as part of course-based undergraduate research experiences (CUREs) (*Hanauer et al., 2017*; *Handelsman et al., 2018*). Citizen scientists, we argue, can also help discover bacteria with novel, useful traits.

*Delftia* is a genus first discovered in the city Delft (*Den Dooren de Jong, 1927*; *Wen et al., 1999*), where bacteria themselves were discovered by Leeuwenhoek (*Gest, 2004*). *Delftia* has genes capable of precipitating gold by excreting a metabolite called delftibactin (*Johnston et al., 2013*). Gold in solution as gold chloride is toxic to bacteria, so *Delftia* has evolved this novel mechanism for precipitating aqueous gold out of solution to nontoxic solid gold nanoparticles. This mechanism has obvious potential uses in gold recycling in used electronics, gold mining, and urban waste (*Gold recycling, 2013*; *Reith et al., 2007*; *Subhabrata, Natarajan & Ting, 2017*), but to date, the existing genetic diversity of *Delftia* in strain collections is modest. There are only six known species of *Delftia*. Full genome assemblies exist for four of these species within the National Center for Biotechnology Information (NCBI) database (*Wen et al., 1999*). Discovery of novel *Delftia* species and their relatives has the potential to better elucidate variations in *Delftia* genetic sequences, especially within the gold precipitation gene cluster and other industrially and human health related sequences. The more information about these gold precipitation genes, for example, the greater potential for using *Delftia* or its genetic potential to recycle our electronics and make mining more sustainable.

Here we leverage a citizen science approach to detect new *Delftia* species on a university campus. We simultaneously test whether students are able to aid the speed of discovery of novel lineages and consider the biology of the lineages we have discovered. The Wolfpack Citizen Science Challenge for spring 2018 (go.ncsu.edu/wpc18) was a

collaborative project to document the presence and genetic diversity of *Delftia* spp. across the North Carolina State University campus and create a *scalable and interdisciplinary* model to continue learning about this and other organisms. In addition to involving students in two introductory courses in the initial data collection, we also involved students in two upper-level courses in the downstream study of the microbes detected during the Challenge.

## MATERIALS AND METHODS

### Recruitment of participants and sample collection

Participants were primarily recruited from two courses, ES 100: Introduction to Environmental Sciences (176 students) and LSC 170: First Year Seminar in the Life Sciences: Meet Your Microbes (20 students). However, anyone interested was able to obtain a sampling kit and participate. A post-event survey indicated that 96% of the participants were required to participate as part of a course and that 48% were currently enrolled as STEM majors.

Three events were held to create excitement and share results from the challenge. In January, the Challenge was launched with a public event attended by 19 people, in which Goller and Riley shared information about *Delftia acidovorans* found in sinks, drains and soil and encouraged members of the campus to think critically about the microbial communities around us. In March, the sequencing data were shared with the campus community at an event at which participants used the NCBI Basic Local Alignment Search Tool (BLAST) to find regions of similarity between the discovered sequences and those deposited in the NCBI database. This BLAST workshop was attended by 55 people. In April, results of the project were shared at a closing event open to the campus and general public, attended by 30 people.

Participants registered as teams of up to five members and were provided kits with instructions and materials to collect samples: three swabs and two 50 mL conical tubes for soil samples along with gloves, plastic spoons for scooping soil, alcohol swabs to sanitize the soil collection spoons and labels for samples. Approximately 40 kits were distributed and over 150 swab and soil samples were received between January 30 and February 14. Samples were delivered in person to either the Biotechnology Program (BIT) teaching laboratories or the NC State University Libraries front desk. Samples were stored in −20 °C freezer until ready for metagenomic DNA extraction. Along with physical samples, metadata including location descriptors and latitude–longitude data were submitted online through a customized SciStarter citizen science website (https://scistarter.com/delftia). Students' identifying information was removed from samples and a numerical identity was assigned.

### Safety

Participants were provided with detailed instructions on how to sample environments around the campus and use the sampling kit. Participants were instructed to use the swab to sample a safe location and immediately place the swab in the transport container. Students collected soil samples with the provided tube and spoon while wearing disposable

gloves. For processing of samples, students in molecular biology courses were trained in lab safety procedures and given a document detailing the potential hazards and safety procedures used in the teaching laboratory. For all extractions and qPCR reactions, students wore provided disposable lab coats, safety glasses and gloves, and disinfected all surfaces before and after use.

## Isolation and purification of metagenomic DNA

Metagenomic DNA was extracted from samples using the Invitrogen PureLink Microbiome DNA Purification Kit according to the corresponding protocol for swab and soil samples. Soil was transferred from collection tubes to bead tubes with alcohol-sterilized metal scoops. Swab tips were cut off into bead tubes with alcohol-sterilized metal scissors. Samples were lysed and homogenized by heat, bead beating and lysis buffer. After purification, samples were eluted in 50 μl of elution buffer. DNA concentration was determined spectrophotometrically using a ThermoFisher NanoDrop 2000c instrument and normalized to five ng/μl. Samples were matched with descriptive location data in an online spreadsheet using information submitted on the SciStarter website (https://scistarter.com/delftia). Isolations were performed by Riley in batches of 12–24 samples.

## Detection of *Delftia*-specific sequences by quantitative real-time PCR

An Eppendorf epMotion 5075 TC liquid handler was used to set up quantitative real-time PCR (qPCR) reactions with New England BioLabs Luna Universal Probe qPCR reagents, primers and double-quenched probes (IDT DNA). qPCR reactions were run on a Bio-Rad CFX Connect instrument and data were exported as spreadsheets with cycle threshold (Ct) values for each reaction. Samples were screened for the quantity of *Delftia* present using double-quenched, *Delftia*-specific primers and probe for a portion of the unique gold biomineralization metabolite production system (hereafter "gold gene"; *Johnston et al., 2013*; GenBank CP000884.1, region 5233319–5234363; see Data S1 for primer and probe sequences, *Seq1, Seq2, Seq3*). Presence and abundance of *Delftia* were then confirmed with a second set of primers and probe for a putative *Delftia*-specific toxin–antitoxin sequence unique to *Delftia* spp. (hereafter "CP sequence"; GenBank CP000884.1, region 759992–760309; see Data S1 for primer and probe sequences, *Seq4, Seq5, Seq6*). Reactions were set up in duplicate along with an 8-point, ten-fold dilution standard curve with "Gold Gene" standard beginning at 40 pg/μl and CP gene standard at 30 pg/μl. Dilution calculation tables and qPCR conditions are described in Data S2.

## Abundance estimation of *Delftia* spp. in samples

Undergraduate juniors and seniors and first- and second-year graduate students enrolled in an upper-level *High-throughput Discovery* 8-week lab module programed an epMotion 5075 TC liquid handler with the qPCR script, prepared metagenomic samples for qPCR and calculated *Delftia* copy numbers using the qPCR Ct data (see Table S1). Students were provided a spreadsheet template with detailed explanations and information on the use of a standard curve for calculation of absolute copy numbers of target sequences. Data were shared with students and groups of three to four were tasked with determining

copy numbers for one 96-well PCR plate containing: 23 genomic DNA samples tested in duplicate along with an 8-point standard curve and negative "buffer only" controls. Multiple groups analyzed the same samples to confirm the results and copy number trends were further supported by analyzing qPCR data for the same samples with a primer set for the single-copy *Delftia*-specific "CP" sequence described above. Data were then analyzed as a class and shared with Danica Lewis (NC State University Libraries) for visualization and dissemination of the results to participants and the public (go.ncsu.edu/exploredelftia). Samples with the highest *Delftia* copy number using both primer sets were selected for further analysis of the unique "gold" sequence.

### Sequencing of "gold gene" in samples positive for *Delftia* spp.

For 20 samples with high *Delftia* counts, a portion of the gold gene sequence was amplified using primers *Seq7* and *Seq8* identified in Data S1 and the Q5(R) High-Fidelity 2X Master Mix (New England Biolabs, Ipswich, MA, USA) according to the protocol outlined in Data S3. The amplified portion of the gold gene was selected because it is highly specific to *Delftia* and based on current sequence database information, varies slightly between known species and strains, allowing for identification from metagenomic samples. The target *Delftia* sequence is 1,045 base pairs in length (Data S4). Of the 20 tested samples, 17 produced sufficient PCR product for sequencing and were sent to the NC State University Genomic Sciences Laboratory (GSL) for Sanger DNA sequencing using primers *Seq7* and *Seq8*. Amplicons were sequenced from both directions and sequences were trimmed based on stringent quality settings to match existing sequences in the NCBI database. The sequencing data were shared with the campus community at an event at which participants used the NCBI Basic Local Alignment Search Tool (BLAST) to find regions of local similarity between the discovered sequences and those deposited in the NCBI database. This allowed participants to identify which *Delftia* species and strains best matched the samples that were sequenced (see Data S5–S9).

### Data dissemination

The Google Maps Fusion Tables extension was used to create a heatmap of *Delftia* presence and abundance across campus and Tableau Public software was used to create an interactive map (http://go.ncsu.edu/ExploreDelftia). Participants were invited to explore the data and evaluate which samples had the highest amount of *Delftia*. Students in the courses involved in sampling and analysis were shown the results and asked to discuss future research questions.

## RESULTS

### Proportion of samples containing *Delftia* spp. sequences

Over 150 samples were received from participants. Of these, 135 were labeled correctly and matched with the online SciStarter database containing sampling location descriptions and latitude–longitude coordinates. Through qPCR analysis using primers and probe *Seq1*, *Seq2*, and *Seq3*, 125 samples (92.6%) had detectable quantities of the target *Delftia* "gold gene" DNA sequence. Quantities of *Delftia* within samples were confirmed using the

**Table 1 Top 20 samples with *Delftia* DNA identified via qPCR targeting of unique "gold gene".**

| DNA Sample number | *Delftia* gold gene count | Latitude | Longitude | Sample type | Location | Description |
|---|---|---|---|---|---|---|
| 15-1 | 113,191 | 35.78593062 | −78.66805315 | Swab | Poe Hall | Water fountain |
| 7-1 | 17,294 | 35.78472956 | −78.67292404 | Swab | Owen Residence Hall | *None provided* |
| 24-1 | 14,167 | 35.78822065 | −78.67522672 | Swab | University Towers | Parking deck drains |
| 1-3 | 12,041 | 35.78654 | −78.671737 | Swab | Williams Hall | Bathroom sink |
| 17-3 | 9,493 | 35.78068018 | −78.67308866 | Swab | Wood Residence Hall | Sink drain |
| 23-2 | 8,780 | 35.78468 | −78.666723 | Swab | SAS building | The girls bathroom sink on the first floor of SAS building, middle sink |
| 33-2 | 5,789 | 35.78744498 | −78.67013454 | Swab | D.H. Hill Jr. Library | 3rd floor women's bathroom sink |
| 12-2 | 5,095 | 35.785385 | −78.673091 | Swab | Metcalf bathroom | Bathroom sink |
| 26-1 | 5,095 | 35.74477072 | −78.68757963 | Swab | Campus Crossing | Apartment complex |
| 9-1 | 4,612 | 35.78795303 | −78.67699295 | Swab | Valentine Commons | Kitchen sink |
| 44-4 | 4,356 | 35.78670028 | −78.67463044 | Soil | Fence on Dan Allen Dr. | Chilly (56 F), drier soil, live organisms present |
| 25-2 | 2,961 | 35.7861221 | −78.66352558 | Swab | NCSU bell tower, main campus | Wild Card sample-seat located on NCSU bell tower |
| 45-4 | 2,317 | 35.78753054 | −78.67083426 | Soil | Atrium | Trash bins next to the vending machines |
| 24-2 | 1,971 | 35.78822065 | −78.67522672 | Swab | University Towers | Drain |
| 15-2 | 1,873 | 35.785982 | −78.677831 | Swab | Lee Hall | Suite 807 Sink |
| 18-1 | 1,733 | 35.78481659 | −78.67285967 | Swab | Owen Residence Hall | Inside in dorm room |
| 25-1 | 1,535 | 35.77153404 | −78.67522001 | Swab | Engineering Building I, Centennial campus | Drain in the middle of the floor of the bathroom |
| 29-2 | 1,452 | 35.78751407 | −78.66981704 | Swab | D.H. Hill Jr. Library | Floor 1 |
| 7-2 | 1,381 | 35.78412031 | −78.67101431 | Swab | Talley Student Union | Bathroom sink drain |
| 30-2 | 1,113 | 35.78824567 | −78.67403984 | Swab | Nelson Hall | Water fountain |

CP qPCR primers and probe *Seq4*, *Seq5* and *Seq6*. The 20 samples with highest *Delftia* counts were primarily swabs from sinks and drains (Table 1). In contrast, the samples with the least *Delftia* DNA tended to be those from soil samples and outdoor locations. However, it is worth reiterating that nearly all of the samples contained some *Delftia*, a relatively understudied genus of bacteria.

We next compared the *Delftia* gold gene sequences in the samples to those of sequenced strains. Collectively, the sequences from our samples were most similar to those of *Delftia tsuruhatensis* strain CM13, *Delftia acidovorans* strains ANG1 and SPH-1, or *Delftia acidovorans* strain RAY209 (see Table 2). Differentiation between *D. acidovorans* strains ANG1 and SPH-1 was not possible as each matched query had the same identity, query coverage and *E* value results for both strains. However, for strains of *D. tsuruhatensis* CM13 and *D. acidovorans* RAY209, the sequences matched with highest probability to each, respectively. None of our samples were close matches for the other sequenced *Delftia* species of *D. deserti*, *D. lacustris*, *D. litopenaei*, *D. rhizosphaerae*, or other strains of

**Table 2 NCBI BLAST results for sequenced environmental "gold gene".**

| Sample | Species and strain | Identity (%) | Query coverage (%) | E value |
|--------|--------------------|--------------|--------------------|---------|
| 1-3 | *Delftia tsuruhatensis* strain CM13 | 98 | 100 | 0.0 |
| 9-1 | | 96 | 100 | 0.0 |
| 12-2 | | 95 | 99 | 0.0 |
| 15-1 | | 97 | 100 | 0.0 |
| 15-2 | | 95 | 100 | 0.0 |
| 17-3 | | 95 | 100 | 0.0 |
| 25-1 | | 95 | 100 | 0.0 |
| 25-2 | | 96 | 100 | 0.0 |
| 26-1 | | 96 | 100 | 0.0 |
| 30-2 | | 95 | 99 | 0.0 |
| 33-2 | | 97 | 100 | 0.0 |
| 7-1 | *Delftia acidovorans* strain ANG1 *or* strain SPH-1 | 91 | 100 | 0.0 |
| 18-1 | | 94 | 99 | 0.0 |
| 23-2 | | 95 | 100 | 0.0 |
| 24-1 | | 95 | 100 | 0.0 |
| 7-2 | *Delftia acidovorans* strain RAY209 | 82 | 99 | 3E−154 |
| 30-3 | | 96 | 99 | 0.0 |

*D. acidovorans* and *D. tsuruhatensis*. A total of 14 out of the 17 sequences had less than 97% sequence identity with the *Delftia* strains they most closely matched.

## DISCUSSION

Here, we sought to simultaneously test whether we could engage students campus-wide in a citizen science style microbial research project and in doing so, understand the distribution and diversity of strains of one particular bacterial genus, *Delftia*. In short, we were indeed able to engage students from diverse majors across campus. In doing so, we discovered that some sampling sites had many more *Delftia* counts than did others, that *Delftia* was relatively ubiquitous and that some of the strains we identified had gold genes that appeared relatively divergent from those known from the literature. Although we were unable to accurately determine the diversity of *Delftia* strains present, this unanswered question presents a new challenge and opportunity for our citizen science and *Delftia* research efforts.

Collectively, the qPCR, Sanger DNA sequencing and BLAST comparison results showed that strains of *Delftia* are diverse, abundant and frequent (found at many sites) in environments in and around the college campus. Based on available genomic sequences deposited in the NCBI database and partial sequencing of the highly conserved gold gene, the strains students discovered best matched the reference strains *D. tsuruhatensis* CM13 and *D. acidovorans* ANG1 and SPH-1. However, 14 of 17 samples contained strains that were a 97% or lower match to strains in the NCBI database. Our suspicion is that these strains represent uncharacterized genetic diversity among strains in *Delftia*'s gold gene.

However, because we sequenced from complex environmental samples we can't preclude the possibility that some of this variation is due to cases in which the forward and reverse sequences obtained were from different *Delftia* species or strains in the sample.

The sequenced *Delftia* gold gene from many of the participant samples matched well to known *Delftia* species, but some samples matched two different existing strains equally well. For example, samples from 7-1 to 24-1 were equally similar to the strains *D. acidovorans* ANG1 and SPH-1 (Table 2). Clearly further work can be done to sequence additional portions or the entire genomes of these samples to identify what known strain is present or discover a new lineage of *Delftia*. More extensive community analyses of the samples using both targeted (16S rRNA gene) and whole genome shotgun sequencing would aid in the identification of which microbes associate with the presence of *Delftia* and the identity of the gold sequences in the environment, respectively. Additionally, high-throughput sequencing approaches such as Hi-C from Phase Genomics ("Hi-C Proximity-Guided Assembly," 2018) (*Sieber et al., 2018*) or Nanopore single-molecule long-read sequencing can be employed to attempt to sequence and assemble the entire *Delftia* genome in metagenomic samples positive for *Delftia* by qPCR. Ultimately, selective media capable of isolating and identifying *Delftia* would allow us to increase our collection of *Delftia* strains for basic functional studies and genome sequencing.

Our sequencing results best matched the species *D. acidovorans* and *D. tsuruhatensis*, both of which have been found in environments similar to those we studied. *D. acidovorans* was originally discovered in soil and has been found in drains, waterspouts and showerheads in the built environment (*Wen et al., 1999*). *D. tsuruhatensis* was first discovered in a wastewater treatment plant and has been found in similar locations along with *D. acidovorans* (*Hou et al., 2015*). The *Delftia* species we did not encounter in our study are species that have so far been associated with more restricted habitats. *Delftia deserti* has been found to inhabit desert environments (*Li et al., 2015*), *D. lacustris* in lake water (*Jørgensen et al., 2009*), *D. litopenaei* in pond water (*Chen et al., 2012*), and *D. rhizosphaerae* in the rhizosphere of *Cistus ladanifer*, a plant native to the Mediterranean region (*Carro et al., 2017*). The apparent ubiquity of the genus *Delftia* hides the reality that individual species appear to show considerable habitat restriction. In the future, it would be interesting to understand which traits and genes of individual *Delftia* species confer the ability to survive in particular habitats.

It is unclear the extent to which the life history of *Delftia* in the above habitats is the same as that of *Delftia* in the built environment of a college campus. Nor is it well understood whether the presence of *Delftia* in water systems is problematic or potentially beneficial. Like many bacterial taxa, *Delftia* species are recorded as opportunistic pathogens that can infect hospitalized or immunocompromised patients (*Patel et al., 2019*) (*Bilgin et al., 2015*). However, there is no indication that human bodies are a common habitat for this genus. Instead, in buildings such as those we sampled it appears to be much more common in water systems—in drains, showerheads and downspouts. In as much as the ecological conditions of water systems differ greatly, it is possible that a comparative study of water systems, such as those that are or are not chlorinated, might reveal more about the built environment natural history of this organism.

Our approach kindled campus-wide student interest in microbial diversity and molecular biology techniques through the excitement of discovering this unique microbe in places that students frequent on campus. Groups of students from various academic disciplines and courses produced and analyzed samples that contributed to a large public dataset. The findings helped teach the student community about *Delftia* and also reinforced the importance of the collaborative nature of scientific discovery. The success of this project, in terms of the documentation of *Delftia*'s distribution helps to validate our general approach. In addition, our approach has the potential to encourage future students to participate. We aim to continue the challenge of accurately identifying new *Delftia* lineages and engage others by expanding the sampling opportunity to a multi-section first-year English class that is required for all undergraduate students on our campus. Using a similar approach and incorporating the expertise of faculty in the English department, we will engage students in writing tasks related to the project. Additionally, an upper-level metagenomics course will tie into this endeavor by processing, sequencing and analyzing the microbial communities in samples with high numbers of *Delftia* sequences. With relatively minor changes to the course schedules and curricula, 100 more students per semester can participate, learn and contribute to the project. We are creating resources that are accessible for other faculty and campuses to implement this project and share findings. For this, students participating in the project are writing the *Delftia* book (go.ncsu.edu/delftiabook), and we have created a group for instructor resources on the QUBES web portal (https://qubeshub.org/community/groups/delftia/projects). Liquid handlers can be cost-prohibitive, but less expensive models such as the Opentrons OT-2 are available, and we are developing scripts for this instrument. Student groups in lab-based courses can always set up qPCRs manually to participate in this project.

## CONCLUSIONS

As the future plans for integrating this project into courses indicate, enthusiasm for the project was high among our colleagues and grew as the project proceeded. However, if we are to continue the project it is key that it continues to yield new scientific insights. Fortunately, this seems very likely to be the case. For example, although *Delftia* abundance was very patchy on campus, we have yet to explain what factors account for such patchiness. Additional samples will help us to have sufficient coverage across sample types to allow spatial models of *Delftia* diversity and abundance. In addition, our results suggest that new variants of the *Delftia* gold gene and even new *Delftia* strains remain to be discovered. Conversely, there is a lack of genomic diversity represented in the NCBI database. By leveraging the enthusiasm of university students and staff, interconnecting courses and researchers, and using our model pipeline, new lineages of *Delftia* can be rapidly identified and studied (e.g., groups of students cloning novel gold gene cluster into a host such as *Escherichia coli* or yeast for functional characterization). This will yield a better understanding of the ecological and environmental significance of these organisms and simultaneously help to connect students and faculty across campus in a common

scientific project. Finally, it is of note that *Delftia* species, while little known, are of potentially great applied importance. In addition, they contain genes that allow many strains to precipitate gold. Given the many waste streams in which gold is present but hard to concentrate, this ability has the potential to be very useful moving forward.

## ACKNOWLEDGEMENTS

We would like to acknowledge the contributions of all the students involved in this project.

### Funding

Financial support was provided by the Biotechnology Program (BIT) at North Carolina State University for the courses and reagents used for this project. This work was supported by the National Science Foundation grant #1319293 (Dr. Rob Dunn).
The funders had no role in study design, data collection and analysis, decision to publish, or preparation of the manuscript.

### Grant Disclosures

The following grant information was disclosed by the authors:
North Carolina State University.
National Science Foundation: #1319293.

### Competing Interests

The authors declare that they have no competing interests.

### Author Contributions

- Noah G. Riley conceived and designed the experiments, performed the experiments, analyzed the data, prepared figures and/or tables, authored or reviewed drafts of the paper, and approved the final draft.
- Carlos C. Goller conceived and designed the experiments, performed the experiments, analyzed the data, prepared figures and/or tables, authored or reviewed drafts of the paper, and approved the final draft.
- Zakiya H. Leggett conceived and designed the experiments, authored or reviewed drafts of the paper, and approved the final draft.
- Danica M. Lewis conceived and designed the experiments, analyzed the data, prepared figures and/or tables, authored or reviewed drafts of the paper, and approved the final draft.
- Karen Ciccone conceived and designed the experiments, authored or reviewed drafts of the paper, and approved the final draft.
- Robert R. Dunn conceived and designed the experiments, analyzed the data, prepared figures and/or tables, authored or reviewed drafts of the paper, and approved the final draft.

## DNA Deposition

The following information was supplied regarding the deposition of DNA sequences:

The raw Sanger sequencing results for the samples with the highest Delftia counts as indicated in Table 2 are available as a Supplemental File.

The sequences and metadata are available in GenBank:

BankIt2283542 Sample_1-3, MN721339;
BankIt2283542 Sample_7-1, MN721340;
BankIt2283542 Sample_7-2, MN721341;
BankIt2283542 Sample_9-1, MN721342;
BankIt2283542 Sample_12-2, MN721343;
BankIt2283542 Sample_15-1, MN721344;
BankIt2283542 Sample_15-2, MN721345;
BankIt2283542 Sample_17-3, MN721346;
BankIt2283542 Sample_18-1, MN721347;
BankIt2283542 Sample_23-2, MN721348;
BankIt2283542 Sample_24-1, MN721349;
BankIt2283542 Sample_25-1, MN721350;
BankIt2283542 Sample_25-2, MN721351;
BankIt2283542 Sample_26-1, MN721352;
BankIt2283542 Sample_30-2, MN721353;
BankIt2283542 Sample_30-3, MN721354;
BankIt2283542 Sample_33-2, MN721355.

## Data Availability

The raw Sanger sequencing results for the samples with the highest *Delftia* counts are available as a Supplemental File.

## Supplemental Information

Supplemental information for this article can be found online at http://dx.doi.org/10.7717/peerj.8925#supplemental-information.

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
