# Peer review of "Catalyzing rapid discovery of gold-precipitating bacterial lineages with university students"

_PeerJ, doi:10.7717/peerj.8925_

## Round 0.1 · original submission · Major Revisions

Three peer-reviewers have provided detailed and informative feedback that I believe will allow you to revise and enhance your article to make it suitable for publication in PeerJ. I recognize that not all of requested additions are feasible (e.g., you may not have performed assessments and can't go back in time) but I think there are several modifications and/or expansions requested that are feasible and would result in an improved manuscript. In the cases where additional information is not possible, please just add brief notes in the discussion regarding caveats of the current study and detail how future iterations could be improved through inclusion of additional analyses.

·

Basic reporting

The authors present an approach to collect a high volume of samples by engaging individuals across an entire campus. The manuscript is well written and concise. There a few instances where the clarity of the manuscript could be enhanced in places. Below are suggested changes:

Line 53: The abstract states there was an analysis of “evolutionary trajectories” however in the study there was only a comparison of ‘gold gene’ sequences obtained in this study to current published sequences. Please remove “evolutionary trajectories”. Suggested replacement is DNA sequences.

Line 62: Add “experimental” before questions.

Line 66-67: The reference listed to support this statement (Schlaberg, Simmon & Fisher, 2012) is a paper that looks at identifying novel pathogens in clinical settings. This doesn’t support studying “unique abilities of bacteria to everyday human life”. Please remove this sentence.

Line 71: change the word trivial (it implies the stated procedure is unimportant)

Lines 75-77: fix indent of sentences

Line 90: Take out “Arguably, one of the most remarkable bacterial taxa is”. While the gold-precipitating ability of this bacteria is a great way to interest the non-science community, some in the scientific community might not find this significant enough to warrant these bacteria as the most remarkable of all bacteria.

Line 93: After obvious potential uses please state a few of the those uses.

Line 98: change taxa to species

Line 111: change ‘classes’ to course to emphasize the ability of the study to be applied to different courses

Line 118: Define CCG and NGR.

Line 132: Italicize Safety

Line 135-136: Reword sentence to include the student was wearing the disposable gloves.

Line 193: please clarify Appendix 4 is a list of all samples.

Line 224-226: Remove this sentence. A quick Pubmed search using “Delftia” results in 373 results. I would not conclude that these are rare bacteria, just not a ‘household name’ bacteria.

Line 243-244: What is meant by neutral? Please clarify this statement.

Experimental design

The goals of this study were to engage the campus community, conceptualize science for students, examine the abundance of Delftia spp. in samples collected and identify novel species of Delftia using DNA sequences of the Delftia-specific gold gene. In my opinion, the authors were highly successful in all their aims except for the last.

Unfortunately, there is a flaw in the latter portion of the experimental design that invalidates some of the conclusions drawn from the results. Since the DNA was extracted from environmental samples it cannot be assumed that only one species of Delftia is represented and therefore sequencing of the gold gene in these samples could be considered ambiguous. The sequences obtained could possibly be a mixture of sequences from various species/strain of Delftia obtained in a single environmental sample. This may explain the areas of your electropherograms that show overlapping peaks. (For example, a quick contig construction for the Forward and Reverse sequencing results for Sample 1-3 shows roughly 8% disagreement of the sequences.)

To elucidate the confounding nature of environmental samples the bacteria should first be isolated or the PCR products should be ligated into a cloning vector and transformed into competent bacterial cells, cloned and plasmids extracted and sent for sequencing.

In addition, after these clean sequences are obtained, it would be beneficial to run a phylogenetic analysis of the sequences obtain in addition to reporting the BLAST results. This would be helpful for future readers to understand the relationship among the reference strains used in the comparison and the environmental isolates. For example, I would like to know how similar the gold gene is in Delftia tsuruhatensis strain CM13 as compared to the gold gene of Delftia acidovorans strain ANG1, SPH-1 and RAY209.

Specific comments:

Line 54 and Line 61: This study did not discover “new lineages” of Delftia spp. It may be more appropriate to say the aim was to assess the prevalence of Delftia spp. based on targeted metagenomic sequencing.

Line 142: Quick note, in future studies you may also want to treat the metal scoops and scissors with DNaway between samples to ensure no free DNA carry over contamination from sample to sample.

Lines 161-168: Please include standard curve information.

Line 165: reference qPCR conditions in the supplemental material

SUPPLEMENTAL 2: The primer and probe concentrations are not listed.

Validity of the findings

The authors present an engaging citizen-science study that shows the power of involving the community in sample collection. The distribution of Delftia spp. and relative abundance is demonstrated using two individual Delftia-specific qPCR reactions. However, some of the specific findings and conclusions made by the authors regarding the discovery of new species cannot be supported by the data due to the complex nature of the DNA sample used as template in the gold gene sequencing reactions.

Specific comments:

Line 239-241: The Schlaberg et al. 2012 study presents species differences based on differences in 16S rRNA genes, this can not be extrapolated to individual genes, more specifically the gene gold.

Lines 251-256: Agreed that there was engagement of students and the campus community that helped to better understand the distribution of Delftia, however the conclusion of new strains of Delftia cannot be stated based on the experimental procedure used in this study.

Line 255-262, 289-290: New species cannot be inferred with certainty based on the issue mentioned above.

Additional comments

Overall, the premise of this paper is very interesting and the manuscript is well written. It is impressive to learn the first author of this paper is an undergraduate researcher and, as a fellow teaching-track faculty, I commend the faculty member on the resourcefulness of recruiting sample collectors outside the classroom. This is a great way to engage the general public and allowed for significant amounts of information to be collected regarding Delftia spp. in a short amount of time, however some of the conclusions made in the manuscript cannot be confidently supported based on the experimental protocol used. However, the authors outline ways to better identify Delftia spp. in their samples in lines 277-287. I suggest removing language from the current manuscript stating the discovery of new strains/species of Delftia and instead focus on the success of the campus-wide initiative and emphasize the presence and abundance of the Delftia in the samples that were collected. (For example, how many of the total samples analyzed for positive for Delftia? Also, is there any research regarding the gold gene copy number in a Delftia cell? This would give an indication of the quantity of Delftia in your samples.)

·

Basic reporting

The manuscript introduction provides a compelling argument for the value of citizen science in helping to identify new species of bacteria by expanding the number and types of sampling sites that can be evaluated. It also provides strong case for focusing on Delftia spp as the target genus because of its unique properties that may be of general interest – the ability to precipitate gold. Overall, the manuscript is well written and some restructuring of the discussion section would be helpful to the reader as some of the information is repetitive and seems to go back and forth between topics. For example, line 262 refers to ten of seventeen samples having less than 95% or lower match and line 291 refers to 14 strains with less than 97% match. Some revision of the discussion section would be helpful for the reader to better understand the impact these results have had on our knowledge about the abundance and distribution of Delftia spp.

Experimental design

The experimental design provides clear instructions on the methods used and are technically appropriate for the project. The unique aspect of the project is the citizen science element and the recruitment of participants and the distribution of sampling kits is critical. The process is reasonably well described and additional information on the public element would be helpful. For example, on line 116 the authors mention a public event to inform the public and recruit participants and it would be helpful to have some more information on how the event was structure and how well attended. Line 120 mentions that participants were primarily recruited from introductory environmental science courses and it would be helpful to know how many participants came from the courses relative to the general public. The manuscript also emphasizes that participants came from a wide-range of disciplines as stated on line 254 and again on line 311 so examples would be helpful. Are the introductory environmental sciences courses non-majors courses that attract students from diverse majors or had these students from diverse majors come from the public event? On line 60, it is mentioned that the results are share through a publicly accessible website and workshops and it would be helpful to have more information about the workshops – how many, how were they structures, how well attended, etc.

Validity of the findings

The results of the project regarding the discovery of potential new species are clear and well documented. The findings are solid and demonstrate the value of citizen science for discovery. The technical aspects are appropriate and readily adaptable to a variety of situations. This illustrates the potential impact of this approach on making new and valuable discoveries. The proposed expansion on these new findings by isolating new Delftia spp. should prove to be very valuable experience for expanding this project into additional courses. The many questions raised regarding the ecology of the organism should lead to opportunities for students to participate in the project and to also have the opportunity to have a critical role in designing the experiments to test the hypothesis that could be generated. I love the future plans to engage additional courses in the project, including a first-year English class. . If available, I would like to see more information about the composition of the participant population to get a better sense of how effectively the project is reaching a broader audience, particularly outside of the sciences.

Additional comments

The manuscript describes a terrific approach to integrate citizen science with course based laboratory projects. It illustrates how an effective program can lead to novel scientific discoveries by engaging the community at large in the sampling process.

Reviewer 3 ·

Basic reporting

This article describes an ambitious effort to engage undergraduate students in citizen science and inquiry-based laboratory modules focused on identification of Delftia in samples from a university campus. This initiative is a valuable contribution to curriculum development in undergraduate science education, and, as discussed in the article, may be adapted to engage non-STEM students. However, in its current state, this article does not present enough data to constitute a substantial contribution to the scholarly literature, and it is not currently suitable for publication in PeerJ.

In the article’s Abstract and Introduction, the authors outline two main goals: (1) “sampling, identification, and analysis of the evolutionary trajectories of a unique yet common microbe, Delftia spp.” with an emphasis on discovery of Delftia lineages able to precipitate gold and (2) enhancing undergraduate learning, specifically focused on “helping contextualize and solidify their learning of the molecular biology techniques”. In its current state, the article does not present enough data to fulfill either of these two goals.

For goal (1) focused on Delftia biology, the authors present data on Delftia presence in 135 samples. Abundance of Delftia in these samples is estimated based on qPCR of two Delftia-specific gene regions. Some analysis of genetic variation of gold precipitation genes is presented, but this is not yet substantial enough to warrant publication. To fulfill goal (1), the authors need more analysis of the gold precipitation gene cluster, expanding to include the entire region. Data are currently presented in tables and alignments, but these data are best presented via phylogenetic analyses and broader comparisons including publicly available database sequences. (For example, are there publicly available metagenomic datasets that could be mined for Delftia sequences?) More data is needed to determine the genetic diversity of Delftia on campus, which is a stated goal in the Introduction. In addition, it is unclear whether the presence of the single 1,045 bp region in the gold precipitation gene cluster is sufficient to conclude whether these Delftia can precipitate gold, which is another stated goal of the article. Overall, although the authors have presented data on sampling and identification of Delftia, there is not enough data on analysis of evolutionary trajectories and genetic variation (which could be achieved by phylogenetic analysis of an expanded gene dataset) or ability to precipitate gold (which could be achieved by analysis of the full gold precipitation gene cluster).

For goal (2) focused on enhancing learning, the authors do not present data on student outcomes. To align with standards in science education research, many inquiry-based curriculum initiatives use pre- and post-assessment instruments to measure student outcomes and determine the impact of the initiative on the targeted student population. I understand that it is not possible to go back in time to conduct these assessments, but this is an important point to consider if the authors are planning to implement this initiative in future courses with the intention of publishing. Even without these assessments, the article in its current state does not present enough data on goal (2) to warrant publication. For example, the article states that students from diverse majors were involved in the project, but there is no data on which majors are represented. One of the aims of the initiative is to “test whether students are able to aid the speed of discovery of novel lineages”, but the outcomes of this aren’t explicitly discussed.

Regarding background/context and literature references, the Introduction lacks a detailed description of the gold precipitation gene cluster in Delftia. Because this is a targeted gene region that is used as an indicator of the presence of Delftia, the authors should present more information on the genes involved in this function and the known variation among published sequences. Considering the model adopted for this initiative, the authors may also want to reference SEA-PHAGES and Tiny Earth (previously the Small World Initiative), which are inquiry-based laboratory and course curricula focused on microbiology.

Experimental design

From the Materials and Methods, it was unclear to me whether all metagenomic DNA was extracted from samples by a single undergraduate researcher (the first author) or by students as part of a course. In addition, it may be worth considering how sample storage and/or processing protocols might affect recovery of Delftia genomic DNA and estimates of abundance.

What were the quality criteria for trimming Sanger sequencing data? Currently, this is described as “stringent quality settings to match existing sequences in the NCBI database”. From the Materials and Methods, it seems that forward and reverse reads were generated. How were these assembled to generate the final consensus sequence? Was manual editing required at any positions?

As mentioned above in Basic Reporting, the analysis of sequence data is currently presented as alignments and tables of BLAST results. These data would be better presented as phylogenetic analyses, which may reveal interesting relationships for further investigation.

Validity of the findings

A major point here involves the authors’ use of sequence similarity cutoffs developed for 16S rRNA gene sequences. As described in the reference cited in this section, the cutoffs of 97% and 95% are relevant for 16S rRNA gene sequences, not necessarily other genes. The 16S rRNA gene has been widely evaluated across many diverse bacterial lineages, and, even so, the cutoffs are often debated among taxonomists. It isn’t appropriate to apply these criteria to other bacterial genes, which experience different selective pressures compared to 16S rRNA genes. With the current dataset, it isn’t possible to estimate the percentage of samples that may contain new Delftia species. It is possible to discuss variation and divergence in gold precipitation genes, which is a more appropriate interpretation of the data.

Additional comments

Overall, in its current state, this article does not present enough data and analysis to achieve the goals outlined by the authors in the Abstract and Introduction. The current manuscript is a solid foundation for a publication, but it is not currently suitable for publication. I have attempted to provide some guidance to the authors regarding further analyses. I understand that budgets and time may be limited, therefore it may not be possible to extend this project beyond the current dataset. If this is the case, the authors may want to consider publishing as a note, with some additional analyses using the existing dataset.

---

## Round 0.2 · accepted · Accept

Thank you for substantially revising the original manuscript based on the reviewers' feedback.